# Non-Alcoholic Steatohepatitis Decreases Microsomal Liver Function in the Absence of Fibrosis

**DOI:** 10.3390/biomedicines8120546

**Published:** 2020-11-27

**Authors:** Wim Verlinden, Eugénie Van Mieghem, Laura Depauw, Thomas Vanwolleghem, Luisa Vonghia, Jonas Weyler, Ann Driessen, Dirk Callens, Laurence Roosens, Eveline Dirinck, An Verrijken, Luc Van Gaal, Sven Francque

**Affiliations:** 1Laboratory of Experimental Medicine and Pediatrics, Division of Gastroenterology and Hepatology, University of Antwerp, 2610 Antwerp, Belgium; eugenie.vanmieghem@gmail.com (E.V.M.); laura.depauw@student.uantwerpen.be (L.D.); thomas.vanwolleghem@uza.be (T.V.); luisa.vonghia@uza.be (L.V.); jonas.weyler@uza.be (J.W.); 2Department of Gastroenterology and Hepatology, Antwerp University Hospital, 2650 Antwerp, Belgium; 3Department of Pathology, Antwerp University Hospital, 2650 Antwerp, Belgium; ann.driessen@uza.be; 4Department of Clinical Biology, Antwerp University Hospital, 2650 Antwerp, Belgium; dirk.callens@uza.be (D.C.); laurence.roosens@uza.be (L.R.); 5Department of Endocrinology, Diabetology and Metabolism, Antwerp University Hospital, 2650 Antwerp, Belgium; eveline.dirinck@uza.be (E.D.); an.verrijken@uza.be (A.V.); luc.vangaal@uza.be (L.V.G.)

**Keywords:** NASH, steatohepatitis, aminopyrine, microsomal, liver function, breath test

## Abstract

The incidence of non-alcoholic fatty liver disease (NAFLD) is rising across the globe, with the presence of steatohepatitis leading to a more aggressive clinical course. Currently, the diagnosis of non-alcoholic steatohepatitis (NASH) is based on histology, though with the high prevalence of NAFLD, a non-invasive method is needed. The ^13^C-aminopyrine breath test (ABT) evaluates the microsomal liver function and could be a potential candidate. We aimed to evaluate a potential change in liver function in NASH patients and to evaluate the diagnostic power of ABT to detect NASH. We performed a retrospective analysis on patients suspected of NAFLD who underwent a liver biopsy and ABT. 440 patients were included. ABT did not decrease in patients with isolated liver steatosis but decreased significantly in the presence of NASH without fibrosis and decreased even further with the presence of significant fibrosis. The predictive power of ABT as a single test for NASH was low but improved in combination with ALT and ultrasonographic steatosis. We conclude that microsomal liver function of patients with NASH is significantly decreased, even in the absence of fibrosis. The ABT is thus a valuable tool in assessing the presence of NASH; and could be used as a supplementary diagnostic tool in clinical practice.

## 1. Introduction

Together with the obesity epidemic, incidences of non-alcoholic fatty liver disease (NAFLD) are rising across the globe. NAFLD is defined as an accumulation of fat in >5% of hepatocytes ocurring in the absence of significant alcohol consumption or any other cause of so-called “secondary” liver steatosis or disease. NAFLD includes two pathologically distinct conditions with different prognoses: isolated steatosis or non-alcoholic fatty liver (NAFL) and non-alcoholic steatohepatitis (NASH), which implies the presence of cell damage and inflammation. The latter covers a wide spectrum of disease severity, including fibrosis, cirrhosis and hepatocellular carcinoma [1].

The ^13^C-aminopyrine breath test (ABT) has been widely used to evaluate microsomal hepatocellular function. Historically, it was the first breath test proposed for the assessment of patients with liver disease and is one of the most frequently utilized and most extensively validated tests for investigating microsomal liver function. The principle of the ABT is based on the selective metabolism of ^13^C-aminopyrine in the liver by the cytochrome P450 mono-oxygenase system of the microsomes [2]. In these microsomes, ^13^C-aminopyrine undergoes 2-step N-demethylation and the appearance of ^13^CO_2_ in breath after administration means that the administered substance underwent microsomal liver oxidation [3,4]. Because N-demethylation of ^13^C-aminopyrine has been shown to be the rate-limiting step, it has been assumed that the ABT reflects the activity of the cytochrome P450-dependent mono-oxygenase system and gives a global assessment of this system. It has been demonstrated that the N-demethylation of aminopyrine is catalyzed most efficiently by CYP2C19 and CYP2C8, followed by CYP2D6, 2C18, 1A2 and 2B6; and to a lesser extent by CYP2C9, 2A6, 1A1 and 3A4 [5,6]. Aminopyrine metabolism is mostly dependent on hepatic metabolic capacity (functional hepatic mass) rather than on portal blood flow [7]. The ABT has been shown to quantitively reflect the severity of cirrhosis, as assessed by other liver function tests [8], and the degree of fibrosis in chronic hepatitis [9,10,11].

Liver function assessment by breath tests based on several metabolic pathways has been studied in the context of NAFLD, mostly to non-invasively diagnose and stage the disease. These studies overall indicate a reduction of liver function in patients with NASH, though included only small groups of patients with significant fibrosis to cirrhosis in the NASH group, making it impossible to differentiate between the effect of NASH and the effect of fibrosis [4,12,13,14,15].

In this study, we aimed to evaluate a change in liver function measured by the ABT in relation to the presence of steatosis, steatohepatitis and fibrosis and the different degrees of severity hereof in a large, prospectively included cohort representing the whole spectrum of disease. Additionally, we aimed to evaluate the diagnostic power of the ABT to detect NASH and fibrosis.

## 2. Methodology

### 2.1. Study Group

We performed a single-center, retrospective study at the Antwerp University Hospital, a tertiary referral center on patient data consecutively collected between 2002 and 2018. Patients visiting the Obesity Clinic or the Hepatology Clinic due to overweight (BMI 25–29.9 kg/m^2^), obesity (BMI ≥ 30 kg/m^2^) or elevated liver enzymes with a suspicion of NAFLD (according to a pre-defined set of criteria) were included when both liver biopsy and ABT were performed. Each patient underwent a standard metabolic work-up combined with a liver-specific program (including ABT as a standard procedure), both approved by the Ethics Committee of the Antwerp University Hospital and requiring written informed consent of the patient (Reference 6/25/15, Belgian Registration Number B30020071389) [16].

### 2.2. Metabolic Work-Up

The metabolic work-up included a detailed questionnaire and a clinical examination with anthropometry. Height was measured to the nearest 0.5 cm and body weight was measured with a digital scale to the nearest 0.2 kg. BMI was calculated as weight in kilograms over height in meters squared. Waist circumference was measured at the mid-level between the lower rib margin and the iliac crest. A blood analysis included blood cell count, coagulation tests, electrolytes, kidney function tests, lipid profile (total and high density lipoprotein (HDL) cholesterols and triglycerides (TG)), liver tests (alanine aminotransferase (ALT), aspartate aminotransferase (AST), gamma glutamyl transpeptidase (GGT), alkaline phosphatase (ALP), total bilirubin and fractions), high-sensitive C-reactive protein (CRP), creatinine kinase, total protein, protein electrophoresis, glucose, insulin and thyroid function.

### 2.3. Hepatological Work-Up

The liver-specific program included additional blood analyses to exclude the classical aetiologies of liver disease (e.g., viral hepatitis and autoimmune disease): s-choline-esterase, carcino-embryonic antigen, α-foetoprotein, anti-nuclear factor, anti-neutrophil cytoplasm antigen antibodies, anti-smooth muscle antibodies, anti-mitochondrial antibodies, anti-liver–kidney microsome antibodies, serum copper and ceruloplasmin, α-1-antitrypsin, anti-hepatitis B core antibodies, hepatitis B surface antigen, anti-hepatitis C virus antibodies. Patients underwent a Doppler ultrasound of the abdomen with parameters of liver and spleen volume and liver vascularization and steatosis grading of the liver based on the Saverymuttu score (ultrasound steatosis, USS, scored 0–3) [17]. USS was scored as: isoechogenicity of the liver and the spleen: 0; slight increase in liver echogenicity, a slight exaggeration of liver and kidney echo discrepancy and relative preservation of echoes from the walls of the portal vein: 1; aforementioned abnormalities accompanied by loss of echoes from the walls of the portal veins, particularly from the peripheral branches, a greater posterior beam attenuation and a greater discrepancy between hepatic and renal echoes: 2; aforementioned abnormalities accompanied by a greater reduction in beam penetration, loss of echoes from most of the portal vein wall, including the main branches, and a large discrepancy between hepatic and renal echoes: 3. Patients also underwent a liver-spleen scintigraphy and an ABT.

### 2.4. Aminopyrine Breath Test

The ABT was carried out at home by the patients at rest after an overnight fasting. The ^13^C-labelled aminopyrine was ingested orally together with water. Aminopyrine is absorbed rapidly and almost completely and breath samples were taken at 0, 30, 60, 90 and 120 min [18]. Peak excretion (ABTpeak) was determined and cumulative excretion (ABTcum) was calculated. Values are expressed as percentage of the administered dose per hour (%dose/h) or the calculated percentage of the administered dose over two hours (%dose/120 min). The analysis of the ABT was performed in the clinical laboratory of the Antwerp University Hospital and was executed by the Automated Breath ^13^C Isotope Ratio Mass Spectrometer (Sercon, Crewe, UK). Normality values of the ABT are >5.4 %dose/h and >8.1 %dose/120 min for peak and cumulative excretion, respectively, based on the available literature and local experience [19].

### 2.5. Liver Biopsy

Liver biopsy was considered indicated in the presence of one or more of the following criteria: persistent abnormal liver tests (AST and/or ALT and/or GGT and/or ALP according to local lab upper limits of normal), ultrasound abnormality of the liver (enlarged liver, steatotic liver [17]), signs of liver disease on liver-spleen scintigraphy [20] and abnormal ABT [21]. A separate informed consent for liver biopsy was required. In patients who subsequently were referred to bariatric surgery, the liver biopsy was performed peri-operatively. The remaining patients were proposed for percutaneous or transjugular liver biopsy. The liver biopsy specimen was stored in formalin aldehyde. Haematoxylin–eosin stain, Sirius red (Fouchet) stain, periodic acid Schiff stain after diastase, reticulin stain (Gordon–Sweets), and Perl’s iron stain were routinely performed on all biopsies and subsequently analysed by a team of experienced pathologists and hepatologists. Biopsies were re-assessed in batch by an experienced pathologist and this reading was used for further analysis. The diagnosis of NASH required the association of some degree of steatosis, some degree of ballooning, and some degree of lobular inflammation [1,22,23]. The different features were scored according to the NASH Clinical Research Network Scoring System [24]. Steatosis was graded as follows: less than 5% of liver parenchyma: 0; 5–33%: 1: 33–66%: 2; more than 66%: 3; Lobular inflammation was scored as: no foci: 0; less than two foci per x200 field: 1; 2–4 foci per x200 field: 2; more than four foci per x200 field: 3. Ballooning was scored as: none: 0; few ballooned cells: 1; many cells/prominent ballooning: 2. Fibrosis was staged: none: 0; perisinusoidal or periportal: 1; perisinusoidal and portal/periportal: 2; bridging fibrosis: 3; cirrhosis: 4. The NAFLD Activity Score (NAS) was calculated as the unweighted sum of the scores for steatosis, ballooning, and lobular inflammation [24]. The length of the biopsy and the number of portal tracts were equally so reported by the pathologist. NASH was defined as the presence of steatosis (≥1), inflammation (≥1) and ballooning (≥1). Borderline NASH was defined as the presence of NASH and a NAS of 3-4. Definite NASH was defined as the presence of NASH and a NAS ≥ 5.

### 2.6. Patient Groups

Patients were excluded from further analysis if they had significant alcohol consumption (>20 g/day for women and >30 g/day for men using self-reported alcohol consumption levels) [25], or if another liver disease was diagnosed. Based on liver biopsy, the study population was divided into different subgroups: patients without signs of steatosis (S = 0) or liver fibrosis: noNAFLD; patients without significant fibrosis (F0–F1), with signs of steatosis (S ≥ 1), and either with the absence of NASH [NAFL (non-alcoholic fatty liver)] or with the presence of NASH (activity and ballooning ≥ 1) (NASH-noF); and patients with significant fibrosis (NAFLD-F). 

### 2.7. Statistical Analysis

The data analyses were performed with SPSS version 25.0 software (IBM Corporation, Armonk, NY, USA). Descriptive statistics were produced for patient characteristics. The distribution of normality was evaluated by the Kolmogorov-Smirnov test and additional visual appraisal of the W-W probability plot. Significant differences in variables between the subgroups were ascertained using independent samples *t*-test for normally distributed continuous variables, the Mann–Whitney U tests for non-normally distributed continuous variables and the Chi-square test for categorical variables. Significant correlations were determined using the Spearman’s rho test. Binary logistic regression analyses were carried out for peak excretion and cumulative excretion of the 13C-ABT. Other variables that were significantly correlated with NASH were included in multivariate logistic regression analyses. A backward elimination method was used to achieve a predictive model for both peak excretion and cumulative excretion separately. Area Under the Receiver Operating Curves (AUC) were generated using the ABT and the predictive models as test variables. AUC values were interpreted as follows: fail (50–60%), poor (60–70%), fair (70–80%), good (80–90%) or excellent (90–100%) [26,27]. Single cut-off values were chosen based on the highest sum of sensitivity and specificity (Youden index). Two-cut off model values were chosen based on 90% specificity and 90% sensitivity. AUROC curves of different tests within the same population were compared according to Delong (MedCalc version 14.12.0, MedCalc Software, Ostend, Belgium) [28]. *p* < 0.05 was considered statistically significant.

## 3. Results

### 3.1. Patient Characteristics

Four hundred and forty patients with a reliable liver biopsy and ABT were included. The total population had an average age of 46.1 years (SD 13.4), median BMI of 37.6 kg/m^2^ (IQR 33.3; 41.7) and a gender distribution of 63.6%/36.4% female to male ratio. All patients (440) were classified according to the different patient groups: noNAFLD (71; 16.1%), NAFL (72; 16.2%), NASH-noF (176; 40.0%) and NAFLD-F (121; 27.5%).

### 3.2. NoNAFLD, NAFL, NASH-noF

Characteristics of the different subgroups are represented in Table 1. There was no statistical difference of the ABTpeak and ABTcum between noNAFLD and NAFL. The ABTpeak and the ABTcum of NASH-noF were, however, significantly lower compared to both the noNAFLD and the NAFL group (Figure 1).

In the NASH-noF group, there was a trend for a decreased ABTpeak and ABTcum between patients with borderline NASH and definite NASH, though these differences did not reach statistical significance [9.20 (6.58; 11.45) vs. 7.65 (5.50; 10.48) %dose/h for ABTpeak (*p* = 0.077); and 12.90 (10.00; 16.55) vs. 10.95 (8.55; 15.13)%dose/120 min for ABTcum (*p* = 0.085), respectively].

In this population, significant positive correlations (Spearman’s rho, *p* < 0.05) were found between ABTpeak and ALT (0.127), LDL cholesterol (0.124), serum albumin (0.182), and smoking (0.282) and significant negative correlations between ABTpeak and BMI (−0.199), waist (−0.157), platelet count (−0.161), steatosis grade (−0.164), lobular inflammation (−0.181), ballooning (−0.136), NAS score (−0.193) and USS (−0.136).

As for ABTcum, significant positive correlations were found with age (0.124), LDL cholesterol (0.132), albumin (0.185) and smoking (0.240) and significant negative correlations between ABTcum and BMI (−0.225), waist (−0.179), platelet count (−0.165), steatosis grade (−0.177), lobular inflammation (−0.195), ballooning (−0.158), NAS score (−0.214), fibrosis stage (−0.123) and USS (−0.145).

The correlation between fibrosis stage and ABTcum in this group without significant fibrosis is most likely due to the confounding effect of NAS score, which was, as might be expected, lower in the NAFL and noNAFLD groups which consisted of less patients with fibrosis stage 1. When analysing this NASH-noF group, no difference of ABT could be observed between patients with fibrosis F0 or F1 [F0 8.4 (6.1–11.6) and F1 8.6 (6.3–10.2), *p* = 0.423; and F0 12.2 (8.9–16.5) and F1 12.1 (8.9–14.7), *p* = 0.535, for ABTpeak and ABTcum respectively]. In this group, there was no correlation between fibrosis stage and ABTpeak (*p* = 0.803) or ABTcum (*p* = 0.833).

### 3.3. Prediction of NASH in Non-Significant Fibrosis

In the combined groups of noNAFLD, NAFL and NASH-noF, a significant correlation (*p* < 0.05) could be found between the presence of NASH and age (0.155), waist (0.156), AST (0.227), ALT (0.268), ABTpeak (−0.174), ABTcum (−0.202) and USS (0.425). When separating ABTpeak and ABTcum, a remaining significant correlation could be found after logistic regression for ALT, USS and ABTpeak; and for ALT, USS and ABTcum. ALT and USS were significantly correlated with the NAS score (*p* < 0.00001, Spearman’s rho 0.314 and 0.557, respectively). USS and ALT were significantly higher in the group of definite NASH compared to the group of borderline NASH.

Based on these data, two models were created to predict the presence of NASH:

PredABTpeak for NASH = −1.617+ (0.025 × ALT) + (0.909 x USS) + (−0.094 × ABTpeak)

PredABTcum for NASH = −1.457 + (0.025 × ALT) + (0.901 × USS) + (−0.075 × ABTcum)

AUROCs for the prediction of NASH and definite NASH for ABTpeak, ABTcum, ALT, USS as single tests and for the predictive models are represented in Table 2. The predictive models were significantly better (*p* < 0.01) than ABTpeak, ABTcum and ALT individually for the prediction of NASH and definite NASH, but not significantly better than USS (*p* > 0.05).

Based on the predictive model of the ABTcum which showed a slightly higher AUROC than ABTpeak, we proposed cut-off values to determine the presence of NASH or definite NASH in Table 3. The predicted model was positively correlated to the presence of NASH. The higher the value, the higher the likelihood of NASH.

### 3.4. Significant Fibrosis

This group consisted of 121 patients with 38.8% F2, 33.9% F3 and 27.3% F4 (i.e., cirrhosis); 28.1% did not have NASH, borderline NASH was present in 19.8% and definite NASH in 52.1% of these cases. In patients with significant fibrosis, ABTpeak and ABTcum are no longer significantly correlated to the presence of NASH (rho −0.104, *p* = 0.258 and rho −0.072, *p* = 0.435, respectively) nor to the NAS score (rho −0.053, *p* = 0.566 and −0.020, *p* = 0.825, respectively). In this population, fibrosis stage has a stronger (inverse) correlation with ABTpeak (rho −0.366, *p* < 0.0001) and ABTcum (rho −0.347, *p* = 0.0001). Fibrosis stage is, however, inversely correlated to the NAS score (rho −0.203, *p* = 0.025).

### 3.5. All Patients

In our entire population, both ABTpeak and ABTcum were independently positively correlated with smoking and serum albumin concentration and negatively correlated with BMI, the presence of NASH and fibrosis stage. Fibrosis was positively correlated with NAS (rho 0.311, *p* < 0.0001) and the presence of NASH (rho 0.222, *p* < 0.0001).

ABTpeak and ABTcum decreased significantly between each fibrosis stage; 9.1 (6.50; 11.80) and 13.40 (9.40; 17.35) for F0–1; 7.80 (6.20; 9.90) and 11.50 (8.80; 14.10) for F2; 6.70 (4.20; 9.40) and 9.50 (6.15; 13.80) for F3; and 3.50 (1.75; 6.80) and 5.00 (2.55; 10.80) for F4, respectively. Differences of ABTpeak and ABTcum for each fibrosis stage are represented in Figure 2.

ABTpeak showed an AUROC of 0.612 (0.559–0.666), 0.674 (0.618–0.731), 0.718 (0.648–0.788) and 0.782 (0.682–0.882) for the prediction of NASH, significant fibrosis, advanced fibrosis and cirrhosis, respectively. ABTcum showed an AUROC of 0.616 (0.0566–0.673), 0.676 (0.619–0.732), 0.719 (0.650–0.788) and 0.769 (0.667–0.872) for the prediction of NASH, significant fibrosis, advanced fibrosis and cirrhosis, respectively.

Based on the ABTcum which showed a slightly higher AUROC than ABTpeak, we proposed cut-off values to determine the presence of significant fibrosis, advanced fibrosis or cirrhosis in Table 3.

## 4. Discussion

The ^13^C-aminopyrine breath test has traditionally been used to estimate functional liver reserve in advanced liver disease. In this study, we demonstrate that non-alcoholic steatohepatitis, even in the absence of significant fibrosis, is associated with a significant impairment of hepatic microsomal function as measured by the ^13^C-aminopyrine breath test. In the fibrotic NAFLD population, the effect of fibrosis on ABT excretion was more important than the effect of inflammation. The ABT can be used as a tool in the prediction of the presence of NASH or fibrosis in the appropriate setting.

Fibrosis has been shown to be the strongest predictor of outcome in NAFLD, which does, however, not equal that it is as such the driver of disease progression [29]. Progression of NALF to bridging fibrosis concurs with transition to NASH and several longitudinal studies have recently confirmed the direct relationship between evolution in disease activity and hepatic inflammatory changes on one hand and the evolution in fibrosis on the other hand. NASH resolution was also shown to be the strongest predictor of fibrosis regression [30,31]. Similarly, SAF activity appeared to be strongly correlated with liver fibrosis stage, further supporting the concept of disease activity as the driving force of disease progression [32,33]. These observations demonstrate that NASH and fibrosis are closely linked.

Multiple serum biomarkers have been evaluated for the prediction of the presence and/or severity of NASH, though most biomarkers failed to demonstrate accuracy. Plasma cytokeratin 18 fragment levels are a marker of hepatocyte apoptosis and represent the most extensively evaluated biomarker of steatohepatitis, although the accuracy is modest. To date, non-invasive tests cannot reliably be used solely for the diagnosis of NASH [34,35]. More extensive research has been performed on non-invasive tools to asses liver fibrosis, though most scores for fibrosis are mainly developed and validated to exclude advanced fibrosis, and do not allow the reliable categorisation of individual liver fibrosis stages [34].

The ABT has mostly been studied in the setting of cirrhosis and end stage liver disease. Smaller studies have previously shown a significant decrease of liver function in NASH patients, measured by isotope breath tests, though these studies were confounded by a limited population size and the inclusion of patients with significant fibrosis to cirrhosis in the NASH groups [4,12,13,14,15].

In this study, we included a large population of patients at risk of NAFLD, by the presence of overweight or obesity, allowing us to investigate the effect of NASH in patients without significant fibrosis and thus isolating the steatohepatitis effect. We observed a significant decrease of microsomal liver function in patients with NASH compared to patients with NAFL and overweight patients without liver steatosis. Liver function was not different between the latter two groups indicating that isolated steatosis does not significantly decrease microsomal liver function. Within the NASH population there was a trend of a decreased ABT excretion between patients with borderline NASH and definite NASH, though these differences did not reach statistical significance. Overall, this indicates that steatohepatitis as such is associated with a significant decrease in microsomal hepatocyte function, which is relevant for our understanding of the pathophysiology of NASH.

Although extraction of aminopyrine seems to be independent of portal blood flow, a role for changes in the microcirculation that have shown to occur early in the development of NASH, even in the non-fibrotic stage, might indirectly play a role [36,37,38,39]. These changes are hypothesised to cause centrolobular hypoxia by aggravating the physiological portocentral oxygen gradient [40]. Hypoxia has been shown to impact on liver microsomal function, so these mechanisms could offer an additional explanation for the presence of microsomal dysfunction early in disease development.

People with overweight or obesity and other components of the metabolic syndrome are at risk of developing fatty liver which is characterized by the presence of large vacuoles of lipids within the cytosol. In addition to macrovacuolar steatosis, NASH implies the presence of cell damage and inflammation and is histologically characterized by microvesicular steatosis, portal and lobular inflammation, and the presence of hepatocyte injury in the form of ballooning and apoptosis. All these processes ultimately stimulate fibrogenesis. At least three major events are involved in the progression of fatty liver to NASH, including overproduction of ROS (reactive oxygen species) and RNS (reactive nitrogen species), lipotoxicity and increased release of proinflammatory and profibrogenic cytokines [41]. These changes impact the microsomal activity and are probably responsible for the decreased microsomal liver function that we observed. Studies with animals and human tissue have also shown an alteration of CYP 450 enzyme activity. CYP3A, a CYP which only plays a minimal role in the ABT, is downregulated, presumably by obesity, elevated proinflammatory cytokines, noncytokine components and oxidative stress [42]. Some CYPs might not only be influenced by the presence of NASH, but could also play a causative role. It has been shown that CYP2E1 (which has no role in ABT) is overexpressed in non-alcoholic steatosis [43]. It is hypothesized that in the case of fat mobilisation as in diabetes mellitus, the hyperketonemia and other small organic molecules are both substrates and inducers of CYP2E1 that will lead to non-alcoholic fatty liver disease. This overexpressed CYP2E1 exhibits a high capacity to produce free radicals that are probably the cause of liver damage and lipid peroxidation in obese type 2 diabetes patients [44].

Since NASH seems to be the driving force for fibrosis development, its diagnosis is paramount, and the ABT could potentially help by discriminating isolated steatosis from NASH. The rising prevalence of NAFLD and the known disadvantages of liver biopsy (sampling error, cost, morbidity and mortality) illustrate the need for new non-invasive diagnostic techniques. Our current results suggest that ABT can indeed be helpful in the differentiation between patients with NASH and those with NAFL in patients without significant fibrosis. This test is non-invasive, innocuous, easy to administer and samples are transportable, which allows its use in primary and secondary care. 

In our group without significant fibrosis, the ABT as a single test had rather poor predictive power, though when including ALT and USS into a predictive model, the predictive power strongly increased. A potential limitation of the use of this model in clinical practice could, however, be hampered by the absence of reliable tools to exclude significant fibrosis. Liver steatosis and ALT have both been related to the presence of NASH. Previous research has shown a correlation between the extent of steatosis (evaluated histologically or ultrasonographically) and the presence of NASH [45,46,47]. In line with our results, Ballestri et al. showed higher USS values in patients with NASH than in those with steatosis; and higher values in patients with definite NASH than in those with borderline NASH [48]. Although normal ALT does not exclude the presence of NASH, studies have shown that ALT levels are independently associated with NASH, even in patients with normal ALT, indicating that even a minor elevation in ALT level, albeit within normal limits, can reflect the presence of NASH-related liver damage [49].

Tribonias et al. found similar results with a substantial impairment of hepatic microsomal function as assessed by a simple non-invasive ABT in NASH patients including, however, patients with significant fibrosis in the NASH group [4]. Compared to our results, they found a higher AUROC [0.741 (0.576–0.905)] for ABTcum to diagnose the presence of NASH, though 25% of their NASH population was cirrhotic compared to 8% in our NASH population. More importantly, 31% of our patients with advanced fibrosis lost the combination of characteristics necessary for the diagnosis of NASH. This is in line with the general observation that with the progression of fibrosis, the characteristic triad of NASH and perisinusoidal fibrosis becomes less prominent or disappears [50].

In our study, we observed, apart from the effect of NASH on microsomal function, a decreasing ABT excretion with each fibrosis stage, which is in line with current literature. Previous reports support this observation and show that ABT results are associated with the severity of liver disease, and that they have a prognostic role in predicting death from liver failure in cirrhotic patients [21,51]. Moreover, we observed that from the moment significant fibrosis is present, the statistically significant influence of NASH on the ABT excretion can no longer be observed.

In our entire population, the predictive power of the cumulative ABT value was mostly higher than the peak ABT value, though not statistically significant. ABT proved to be a poor predictor of significant fibrosis as a single test with an AUROC of 0.676 for ABTcum. The power increased to a fair test to predict the presence of advanced fibrosis and cirrhosis with an AUROC of 0.719 and 0.769 for ABTcum, respectively.

Our findings clearly indicate that both steatohepatitis and fibrosis are associated with impairment of microsomal function. This implies that both these aspects should be taken into account when assessing the accuracy of the ABT as a non-invasive marker of disease. The mixing-up of both aspects may in part explain overall low accuracy and conflicting results in the literature.

ABT is not the only breath test used to investigate hepatic function. ^13^C breath tests can explore either microsomal, cytosolic or mitochondrial hepatocellular subfunctions [2]. Previous research has shown that these other ^13^C breath tests are also capable of distinguishing patients with various degrees of liver disease from normal subjects, as well as distinguishing patients with compensated cirrhosis from those with decompensated cirrhosis. Banasch et al. used a ^13^C-methionine breath test to evaluate mitochondrial dysfunction. They showed in patients without significant fibrosis a difference of mitochondrial function between borderline and definite NASH, which was no longer present in patients with significant fibrosis [52]. In the borderline NASH group, however, 15% had significant fibrosis compared to 43% in the definite NASH group (and only 9% in the NAFL group). Correlation analyses confirmed the synergistic negative effect of NASH activity and fibrosis on individual breath test outcome. NASH activity and fibrosis were correlated as was observed in our study.

Miele et al. performed ^13^C-octanoate breath tests (OBT) in patients with NAFLD and demonstrated the relationship between the presence of fibrosis and the impairment of liver function, expressed by lower OBT results than those of controls [53]. Park et al. showed that the ^13^C-caffeine breath test, another test for microsomal function, reflected the extent of hepatic fibrosis in NALFD and was an independent predictor of significant fibrosis in these patients. They showed impaired liver function in NASH patients, but induced by the presence of significant fibrosis, as they found no correlation with steatosis or inflammation [14].

In our study, ABT results were found to have an inverse correlation with BMI, which is consistent with previous findings as increasing BMI has been repeatedly shown to be associated with worse liver histologic lesions in the NAFLD patients [4,54].

A weakness of our study is the fact that the majority of patients were obese, but the sample included relatively few diabetic patients. The generalizability of the findings to NAFLD patients with a different metabolic profile hence needs to be studied. Furthermore, there is the confounding effect of patient characteristics on the ABT and the cross-sectional nature of the study. Enzymatic functions explored by the ABT may be influenced by hormones, malnutrition, heart or renal failure, sex or xenobiotics such as medication or smoking [2]. No data concerning menstrual cycle or exogenous female sex hormones were obtained during our diagnostic work-up. Smoking induces CYP450 enzymes (primarily CYP1A2); hence a higher peak and cumulative excretion can be expected in smokers [55]. In our non-fibrotic population, smoking was independently correlated with ABT values, though smoking habits did not differ between our subpopulations. In the literature, older age and female gender are correlated with lower ABT values, which could not be observed in our population [56]. Genetic polymorphisms in CYPs are a major cause of the inter individual variation in drug metabolism. Several of the CYPs important for the aminopyrine metabolisation (such as CYP2C19, 2D6, 1A2, 2C9, 2A6) are known to be functionally polymorphic [44]. Rapid metabolisers will have higher ABT results compared to slow metabolisers. In this study, determination of CYP polymorphisms was not performed. Most likely it will not have influenced our findings due to the large subgroups. The strength of our study is first of all the large patient population, which allows us to compare groups independent from the impact of significant fibrosis and which decreases the confounding effect of ABT influencers in individual patients. Furthermore, a large proportion of the patients came in for a problem of overweight or obesity and underwent a liver assessment without a priori suspicion of liver disease. This implies that all patients underwent ABT, and not only those selected because of elevated liver tests or other indicators of liver disease, avoiding a lot of potential bias of which many biopsy-proven NAFLD cohort studies suffer. This methodological approach more closely resembles the context of use in which biomarkers for NAFLD will be used in the future and reinforces the validity of these results and their relevance for routine clinical practice.

To confirm our results, more studies should be performed in large patient groups without the confounding factor of significant or advanced fibrosis. Other compounds than aminopyrine can be used to assess the decreased microsomal function in NASH patients, such as caffeine, phenacetin or methacetin.

The observations in this study open new possibilities for the use of the ABT in NAFLD. The ABT can be incorporated in the diagnostic toolset for NASH (in our model, together with ALT and USS) or fibrosis assessment. A decreased microsomal liver function in patients with NASH might indicate a decreased metabolism of xenobiotics, which has a major impact considering the high global prevalence of NASH. Future studies of NASH medication should take this into account. The ABT could be used to monitor disease progression of inflammation in the absence of fibrosis. Furthermore, the ABT could be an additional tool the evaluate the potential improvement upon (new) therapeutic interventions. 

## 5. Conclusions

The present study shows a decreased microsomal function in NASH patients without significant fibrosis compared to patients with simple steatosis, as assessed by ABT. Fibrosis, when present, has a stronger impact on ABT excretion compared to inflammation and decreases ABT excretion with each fibrosis stage. ABT might be a useful tool for the decision to conduct a liver biopsy in the NAFLD patients and potentially also for the prospective monitoring of disease progression or of the potential benefits after therapeutic interventions.

## Figures and Tables

**Figure 1 biomedicines-08-00546-f001:**
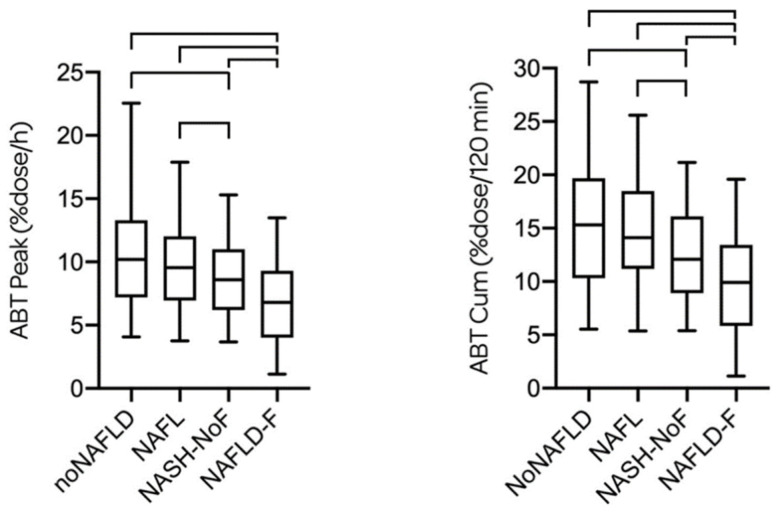
ABTpeak and ABTcum are given for the four subgroups. There was no statistical difference of the ABTpeak and ABTcum between noNAFLD [10.20 (7.20; 13.30) %dose/h and 15.30 (10.30; 19.70) %dose/120 min, respectively] and NAFL [9.55 (6.95; 12.03) %dose/h and 14.10 (11.20; 18.48) %dose/120 min, respectively]. The ABTpeak and the ABTcum of NASH-noF [8.60 (6.20; 11.00) %dose/h and 12.10 (8.90; 16.10) %dose/120 min, respectively] were significantly lower compared to both the noNAFLD and the NAFL group. The ABTpeak and the ABTcum of NAFLD-F [6.80 (4.00; 9.30) %dose/h and 9.90 (5.85; 13.45) %dose/120 min, respectively] were significantly lower compared to all three other subgroups.

**Figure 2 biomedicines-08-00546-f002:**
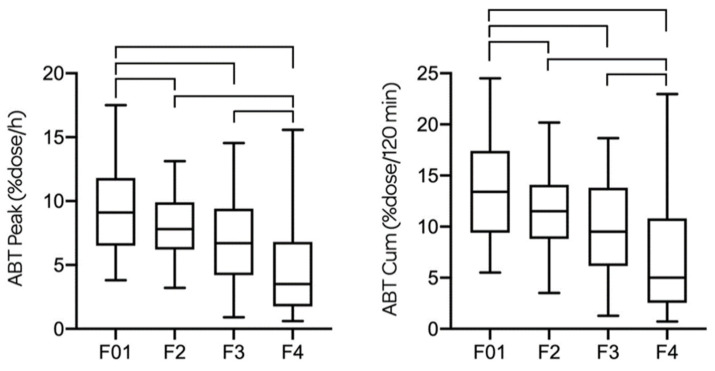
ABTpeak and ABTcum are given for every fibrosis stage in all patients. There is a significant difference of ABTpeak between F2–F4 (*p* < 0.0001) and F3–F4 (*p* = 0.008), but not between F2–F3 (*p* = 0.133). There is a significant difference of ABTcum between F2–F4 (*p* < 0.0001) and F3–F4 (*p* = 0.013), but not between F2–F3 (*p* = 0.119). The ABTpeak and ABTcum are both significantly higher for F0–1 compared to F2 (*p* < 0.05), F3 (*p* < 0.001) and F4 (*p* < 0.0001).

**Table 1 biomedicines-08-00546-t001:** Patient characteristics.

Characteristic	NoNAFLD	NAFL	NASH-noF	NAFLD-F	*p*	*p*	*p*
	(*n* = 71)	(*n* = 72)	(*n* = 176)	(*n* = 121)	noNAFLD vs. NAFL	noNAFLD vs. NASH-noF	NAFL vs. NASH-noF
Age (yrs)	41.23 ± 12.19	45.35 ± 12.49	45.69 ± 12.83	50.03 ± 14.26	0.048 *	0.013 *	0.846
Gender (female)	60 (84.5%)	49 (68.1%)	111 (63.1%)	60 (49.6%)	0.021 *	0.001 *	0.456
Smoking (non-smoker)	52 (73.2%)	55 (76.4%)	139 (79.0%)	89 (73.4%)	0.659	0.259	0.554
BMI (kg/m^2^)	36.80 (33.00; 40.30)	37.77 (32.61; 41.40)	37.70 (33.70; 41.80)	38.61 (32.97; 42.85)	0.474	0.187	0.724
Waist (cm)	111.0 ± 10.7	116.1 ± 15.9	116.9 ± 12.5	124.1 ± 14.6	0.023 *	<0.001 *	0.682
AST (U/L)	24.0 (21.0; 29.5)	26.0 (21.0; 29.8)	29.0 (24.0; 37.0)	37.0 (27.0; 52.0)	0.268	<0.001 *	0.007 *
ALT (U/L)	31.0 (24.0; 42.0)	35.0 (29.3; 46.0)	43.0 (32.0; 57.0)	48.0 (35.0; 75.0)	0.018 *	<0.00001 *	0.007 *
GGT (U/L)	30 (21; 49)	32 (26; 49)	36 (27; 49)	50 (33; 117)	0.13	0.026 *	0.47
PLT (10^9^/L)	282 (252; 339)	268 (232; 323)	290 (246; 329)	249 (177; 302)	0.187	0.92	0.219
Tot chol (mg/dL)	201 (170; 226)	208 (176; 228)	203 (177; 227)	185 (160; 216)	0.805	0.717	0.905
HDL (mg/dL)	50 (42; 65)	47 (39; 56)	46 (39; 55)	41 (35; 51)	0.041 *	0.010 *	0.993
TG (mg/dL)	118 (84; 165)	135 (104; 204)	147 (105; 213)	148 (102; 208)	0.051	0.002 *	0.433
LDL (mg/dL)	118 (94; 142)	125 (98; 151)	120 (100; 146)	114 (82; 139)	0.632	0.672	0.911
Tot bili (mg/dL)	0.50 (0.40; 0.60)	0.50 (0.40; 0.70)	0.50 (0.40; 0.70)	0.60 (0.41; 0.80)	0.333	0.047 *	0.473
HbA1c (%)	5.50 (5.30; 5.70)	5.50 (5.30; 5.80)	5.60 (5.30; 5.90)	5.90 (5.50; 6.90)	0.048 *	0.011 *	0.661
Alb (g/dL)	4.37 ± 0.37	4.41 ± 0.37	4.46 ± 0.41	4.21 ± 0.72	0.539	0.097	0.333
INR	1.00 (1.00; 1.03)	1.01 (1.00; 1.04)	1.02 (1.00; 1.05)	1.04 (1.00; 1.12)	0.537	0.205	0.56
Steatosis					<0.00001 *	<0.00001 *	<0.00001 *
0	71 (100%)	0	0	12 (9.9%)			
1	0	60 (83.3%)	67 (38.1%)	36 (29.8%)			
2	0	11 (15.3%)	68 (38.6%)	37 (30.6%)			
3	0	1 (1.4%)	41 (23.3%)	36 (29.8%)			
Inflammation					0.204	<0.00001 *	<0.00001 *
0	59 (83.1%)	53 (73.6%)	0	24 (19.8%)			
1	12 (16.9%	17 (23.6%)	115 (65.3%)	56 (46.3%)			
2	0	2 (2.8%)	46 (26.1%)	26 (21.5%)			
3	0	0	15 (8.5%)	15 (12.4%)			
Ballooning					0.003 *	<0.00001 *	<0.00001 *
0	60 (84.5%)	43 (59.7%)	0	16 (13.2%)			
1	10 (14.1%)	22 (30.6%)	92 (52.3%)	48 (39.7%)			
2	1 (1.4%)	7 (9.7%)	84 (47.7%)	57 (47.1%)			
NAS					<0.00001 *	<0.00001 *	<0.00001 *
0	49 (69.0%)	0	0	6 (5.0%)			
1	19 (26.8%)	21 (29.2%)	0	5 (4.1%)			
2	2 (2.8%)	36 (50.0%)	0	9 (7.4%)			
3	1 (1.4%)	13 (18.0%)	31 (17.6%)	18 (14.9%)			
4	0	2 (2.8%)	51 (29.0%)	20 (16.5%)			
5	0		43 (24.4%)	25 (20.7%)			
6	0		33 (18.8%)	20 (16.5%)			
7	0		15 (8.5%)	13 (10.7%)			
8	0		3 (1.7%)	5 (4.1%)			
USS					<0.00001 *	<0.00001 *	0.001 *
0	25 (35.2%)	4 (5.6%)	6 (3.4%)	8 (6.6%)			
1	31 (43.7%)	28 (38.9%)	30 (17.0%)	25 (20.7%)			
2	11 (15.5%)	20 (27.8%)	52 (29.5%)	26 (21.5%)			
3	4 (5.6%)	18 (25.0%)	80 (45.5%)	52 (43.0%)			
missing	0	2 (2.8%)	8 (4.5%)	10 (8.3%)			
ABTpeak	10.20 (7.20; 13.30)	9.55 (6.95; 12.03)	8.60 (6.20; 11.00)	6.80 (4.00; 9.30)	0.423	0.005 *	0.031 *
ABTcum	15.30 (10.30; 19.70)	14.10 (11.20; 18.48)	12.10 (8.90; 16.10)	9.90 (5.85; 13.45)	0.535	0.002 *	0.007 *

Patient characteristics of the subgroups: patients without NAFLD (noNAFLD), patients with non-alcoholic fatty liver (NAFL), patients with non-alcoholic steatohepatitis without significant fibrosis (NASH-noF) and patients with significant fibrosis (NAFLD-F). There was a significant difference between NAFLD-F and all three other subgroups for age, gender, waist circumference, AST, ALT, GGT, platelet count, total cholesterol, HDL, bilirubin, HbA1c, serum albumin, INR, steatosis, inflammation, ballooning, NAS, ABTpeak and ABTcum (full *p* values are given as Appendix A). LDL was significantly lower and TG higher in NAFLD-F compared to NASH-noF and noNAFLD, respectively. USS was significantly higher in NAFLD-F compared to noNAFLD and NAFL. Data are expressed as mean ± SD for normally distributed variables or as median (interquartile range) when distribution of the variable is skewed. *p*-value is calculated between different groups with * indicating statistical significance (<0.05). BMI, body mass index; AST, aspartate aminotransferase; ALT, alanine aminotransferase; GGT, gamma glutamyl transpeptidase; PLT, platelets; tot chol, total cholesterol; HDL, high density lipoprotein cholesterols; TG, triglycerides; LDL, low density lipoprotein cholesterols; tot bili, total bilirubin; HbA1c, haemoglobin A1c; Alb: albumin; INR, international normalized ratio; NAS, NAFLD activity score; USS, ultrasound steatosis score; ABTpeak, aminopyrine breath test peak value; ABTcum, aminopyrine breath test cumulative value.

**Table 2 biomedicines-08-00546-t002:** AUROC to predict the presence of NASH and definite NASH.

Test	NASH	Definite NASH
ABTpeak	0.601 (0.539–0.663)	0.612 (0.544–0.681)
ABTcum	0.617 (0.555–0.679)	0.620 (0.553–0.687)
ALT	0.655 (0.595–0.715)	0.669 (0.604–0.734)
USS	0.744 (0.688–0.800)	0.772 (0.717–0.827)
PredABTpeak	0.785 (0.734–0.835)	0.814 (0.765–0.863)
PredABTcum	0.787 (0.737–0.837)	0.819 (0.770–0.867)

AUROC (represented with 95% confidence interval) to predict the presence of NASH and definite NASH for ABTpeak, ABTcum, ALT and US steatosis separately, and the predictive models PredABTpeak and PredABTcum combining ALT, USS and ABT.

**Table 3 biomedicines-08-00546-t003:** Cut-off values to predict the presence of NASH and fibrosis.

	Cut-Off Value	Sens	Spec	PPV	NPV
NASH	0.2575	72.60%	71.60%	75.30%	68.70%
Def NASH	0.2575	87.60%	61.80%	38.10%	92.50%
Sign F	13.25	73.60%	51.40%	36.50%	83.70%
Adv F	10.1	63.50%	69.90%	30.00%	90.50%
Cirrhosis	5.05	51.50%	95.30%	52.80%	96.00%
NASH	>1.1210			82.90%	
No NASH	<−0.6185			80.00%	
Def NASH	>1.1346			64.10%	
No def NASH	<0.0120			93.30%	
Sign F	<7.05			53.70%	
No sign F	>17.45			86.50%	
Adv F	<7.05			46.30%	
No adv F	>18.25			90.70%	
Cirrhosis	<6.55			30.50%	
No cirrhosis	>19.45			94.70%	

Cut-off values to predict the presence of NASH in patients without significant fibrosis; and to predict fibrosis in the whole population with sensitivity, specificity, positive predictive value and negative predictive value. One and two cut-off models were created for each parameter. Using 1 cut-off, the PPV for definite NASH was low (78/162, 38.1%), but of those misclassified as definite NASH, 52% had borderline NASH. Using 2 cut-offs, the PPV for definite NASH was better (41/64, 64.1%), and with 61% of those misclassified having borderline NASH. Using two cut-off values can lead to patients with an indeterminate value. This “indeterminate” classification was present in 54%, 36%, 65%, 68% and 74% of cases for NASH, definite NASH, significant fibrosis, advanced fibrosis and cirrhosis, respectively. Def, definite NASH; Sign F, significant fibrosis; Adv F, advanced fibrosis; Sens, sensitivity; Spec, specificity; PPV, positive predictive value; NPV, negative predictive value.

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
