# Peer review of "Non-Alcoholic Steatohepatitis Decreases Microsomal Liver Function in the Absence of Fibrosis"

_biomedicines, 2020, doi:10.3390/biomedicines8120546_

Round 1
Reviewer 1 Report
The authors have aimed to evaluate a potential change in liver function in NASH patients and to evaluate the diagnostic power of ABT to detect NASH. The authors performed a retrospective analysis on 440 patients suspected of NAFLD who underwent a liver biopsy and ABT and found that ABT did not decrease in patients with isolated liver steatosis but decreased significantly in the presence of NASH without fibrosis and decreased even further with the presence of significant fibrosis. The predictive power of ABT as a single test for NASH was low but improved in combination with ALT and ultrasonographic steatosis. Therefore, the ABT can be a valuable tool in assessing the presence of NASH; and could be used as a supplementary diagnostic tool in clinical practice. The manuscript is well written and my concern is listed below:
1. The Authors mentioned in the discussion that other 13C breath tests are also capable of distinguishing patients with various degrees of liver disease from normal subjects, as well as distinguishing patients with compensated cirrhosis from those with decompensated cirrhosis. Then, what was the reason for choosing 13C-aminopyrine breath tests in this study?
Reviewer 2 Report
This submission is designed to evaluate the diagnostic power of 13C-aminopyrine breath test (ABT) to detect the non-alcoholic steatohepatitis (NASH). I like to give the following comments.
- The degree of fibrosis seems widely used the 13 C-caffeine breath test. ABT applied reference(s) need to add.
- Area under the Receiver Operating Curves (AUC) needs the reference to support.
- NASH and NAFLD must distinguish in clear. Particularly, some plasma biomarkers were not analyzed. Why?
- Aminopyrine metabolism is the key way of ABT. It seems not related to microsomal regulation. Please revise the title of this report.
- ALT and USS may enhance the application of ABT. Why? Please add it in the discussion.
- ABT excretion is reduced with fibrosis depending on the stage. It seems not consistent with the current study. Why?
- ABT seems to show an inverse correlation with BMI. But this novel view did not conduct in detail.
- In conclusion, a decreased microsomal function in patients with NASH that needs more data to support the reduced microsomal function.
- Development of ABT for fibrosis must compare with others such as 13C Octanoate breath tests (OBT) or 13C Caffeine breath test.
Author Response
Response to Reviewer 2 Comments
This submission is designed to evaluate the diagnostic power of 13C-aminopyrine breath test (ABT) to detect the non-alcoholic steatohepatitis (NASH). I like to give the following comments.
Point 1: The degree of fibrosis seems widely used the 13 C-caffeine breath test. ABT applied reference(s) need to add
Response 1: two extra references were added to the two references on the relationship between the ABT and liver fibrosis (p 2 line 12):
Reference 10: Giannini, E.; Fasoli, A.; Chiarbonello, B.; Malfatti, F.; Romagnoli, P.; Botta, F.; Testa, E.; Polegato, S.; Fumagalli, A.; Testa, R. 13C-aminopyrine breath test to evaluate severity of disease in patients with chronic hepatitis C virus infection. Aliment. Pharmacol. Ther. 2002, 16, 717–725, doi:10.1046/j.1365-2036.2002.01200.x.
Reference 11 : Herold, C.; Heinz, R.; Niedobitek, G.; Schneider, T.; Hahn, E.G.; Schuppan, D. Quantitative testing of liver function in relation to fibrosis in patients with chronic hepatitis B and C. Liver 2001, 21, 260–265, doi:10.1034/j.1600-0676.2001.021004260.x.
Point 2: Area under the Receiver Operating Curves (AUC) needs the reference to support.
Response 2: Two references were added to support the use of AUC and the interpretation of the values. P4 line 23
Reference 27. Tape, T.G. Interpreting diagnostic tests: the area under an ROC curve. Univ. Nebraska Med. Cent., doi:http://gim.unmc.edu/dxtests/ROC3.htm.
Reference 28. Mandrekar, J.N. Receiver operating characteristic curve in diagnostic test assessment. J. Thorac. Oncol. 2010, 5, 1315–1316, doi:10.1097/JTO.0b013e3181ec173d.
Point 3: NASH and NAFLD must distinguish in clear. Particularly, some plasma biomarkers were not analyzed. Why?
Response 3: We tried to explain the difference between NASH and NAFLD in our article on P1 line 37-39, P 3 line 44-47 and P10 line 3-5.
It was not the scope of this study to evaluate biomarkers for the diagnosis of NASH. The use of biomarkers is described in the conclusion on P 8 line 24-27. Instead, we opted for the gold standard of liver biopsy to subclassify our patients into noNAFLD, NAFL, NASH-NoF and NAFLD-F, eliminating the need to use biomarkers.
Point 4: Aminopyrine metabolism is the key way of ABT. It seems not related to microsomal regulation. Please revise the title of this report.
Response 4:
Depending on the test compound administered during a breath test, different metabolic pathways (microsomal, cytosolic, mitochondrial) can be examined. The microsomal liver function can be represented using compounds such as aminopyrine, phenacetin, methacetin or caffeine, whereas cytosolic function can be represented by galactose or phenylalanine and mitochondrial function by alfa-ketoisocaproic acid and methionine.
Because generally in literature, the ABT is used to represent the microsomal hepatocellular function, we chose this title.
In the discussion on P11 line 14-15 we mention the testing of mitochondrial and cytosolic testing. A reference was now added to this line:
Reference 2: Armuzzi, A.; Candelli, M.; Zocco, M.A.; Andreoli, A.; De Lorenzo, A.; Nista, E.C.; Miele, L.; Cremonini, F.; Cazzato, I.A.; Grieco, A.; et al. Review article: Breath testing for human liver function assessment. Aliment. Pharmacol. Ther. 2002, 16, 1977–1996, doi:10.1046/j.1365-2036.2002.01374.x.
Point 5: ALT and USS may enhance the application of ABT. Why? Please add it in the discussion.
Answer 5: Thank you for this comment. I added the following text to Results, Discussion and Supplementary table:
P7 lines 6-8 Results:
ALT and USS were significantly correlated with the NAS score (p<0.00001, Spearman's rho 0.314 and 0.557, respectively). USS and ALT were significantly higher in the group of definite NASH compared to the group of borderline NASH.
P5 table 1:
P values were added to the table for steatosis, inflammation, ballooning, NAS and USS using Chi-Square test.
P12 Supplementary table 1:
P values were added to the table for steatosis, inflammation, ballooning, NAS and USS using Chi-Square test.
P10 line 31-39 Discussion:
Liver steatosis and ALT have both been related to the presence of NASH. Previous research has shown a correlation between the extent of steatosis (evaluated histologically or ultrasonographically) and the presence of NASH [46–48]. In line with our results, Ballestri et al showed higher USS values in patients with NASH than in those with steatosis; and higher values in patients with definite NASH than in those with borderline NASH [49]. Although normal ALT does not exclude the presence of NASH, studies have shown that ALT levels are independently associated with NASH, even in patients with normal ALT, indicating that even a minor elevation in ALT level, albeit within normal limits, can reflect the presence of NASH-related liver damage [50].
46 Zardi, E.M.; De Sio, I.; Ghittoni, G.; Sadun, B.; Palmentieri, B.; Roselli, P.; Persico, M.; Caturelli, E. Which clinical and sonographic parameters may be useful to discriminate NASH from steatosis? J. Clin. Gastroenterol. 2011, 45, 59–63, doi:10.1097/MCG.0b013e3181dc25e3.
- Chalasani, N.; Wilson, L.; Kleiner, D.E.; Cummings, O.W.; Brunt, E.M.; Ünalp, A. Relationship of steatosis grade and zonal location to histological features of steatohepatitis in adult patients with non-alcoholic fatty liver disease. J Hepatol 2008, 48, 829–834, doi:10.1016/j.jhep.2008.01.016.
- Liang, R.J.; Wang, H.H.; Lee, W.J.; Liew, P.L.; Lin, J.T.; Wu, M.S. Diagnostic value of ultrasonographic examination for nonalcoholic steatohepatitis in morbidly obese patients undergoing laparoscopic bariatric surgery. Obes. Surg. 2007, 17, 45–56, doi:10.1007/s11695-007-9005-6.
- Ballestri, S.; Lonardo, A.; Romagnoli, D.; Carulli, L.; Losi, L.; Day, C.P.; Loria, P. Ultrasonographic fatty liver indicator, a novel score which rules out NASH and is correlated with metabolic parameters in NAFLD. Liver Int. 2012, 32, 1242–1252, doi:10.1111/j.1478-3231.2012.02804.x.
- Fracanzani, A.L.; Valenti, L.; Bugianesi, E.; Andreoletti, M.; Colli, A.; Vanni, E.; Bertelli, C.; Fatta, E.; Bignamini, D.; Marchesini, G.; et al. Risk of severe liver disease in nonalcoholic fatty liver disease with normal aminotransferase levels: a role for insulin resistance and diabetes. Hepatology 2008, 48, 792–798, doi:10.1002/hep.22429.
Point 6: ABT excretion is reduced with fibrosis depending on the stage. It seems not consistent with the current study. Why?
Response 6: It is indeed known that ABT excretion is reduced depending on the fibrosis stage. The current study also shows a decreased ABT excretion with each fibrosis stage as can be seen on figure 2 and P8 line 25-28. This is in accordance with most of the literature, as is discussed in the discussion on p 10, line 50-51.
Point 7: ABT seems to show an inverse correlation with BMI. But this novel view did not conduct in detail.
Response 7: The inverse correlation of ABT and BMI is not novel and has been observed previously. This is discussed on P11 line 33-35
Point 8: In conclusion, a decreased microsomal function in patients with NASH that needs more data to support the reduced microsomal function.
Response 8: This is indeed a novel result that is shown in our study by including a large patient group allowing us to exclude the confounding effect of fibrosis. More data are indeed needed.
The following text has been added at the end of the discussion:
3To confirm our results, more studies should be performed in large patient groups without the confounding factor of significant or advanced fibrosis. Other compounds than aminopyrine can be used to assess the decreased microsomal function in NASH patients, such as caffeine, phenacetin or methacetin.3
Point 9: Development of ABT for fibrosis must compare with others such as 13C Octanoate breath tests (OBT) or 13C Caffeine breath test.
Answer 9: The comparison with the OBT and Caffeine Breath test were mentioned on P 11 line 25-31. The novelty of this study was the decrease of the ABT excretion induced by inflammation. The effect of fibrosis on ABT has already been demonstrated. We estimated that further description of the correlation of other breath tests and fibrosis was out of the scope of this study and was not further elaborated upon.
Further elaboration could be added if requested, such as:
Schmilovitz et al showed that the caffeine breath test can reliably predict significant hepatic fibrosis in patients with NAFLD with an AUROC of 0.788, which is higher compared to our results (0.674). This difference could be explained by the fibrosis distribution with our population having a smaller proportion of patients with significant fibrosis (27.5% vs. 69.2%) [Ref X1]. Fierbinteanu et al evaluated the use of the 13C-methacetin breath test, another microsomal liver function test, in a population of NASH, NAFL and healthy controls and found excellent results to predict the presence of significant fibrosis, advanced fibrosis and cirrhosis, with AUROC of 0.900, 0.936 and 0.973, respectively [Ref X2].
[Reference X1] Schmilovitz-Weiss H, Niv Y, Halpern M, Sulkes J, Braun M, Barak N, et al. The 13C-caffeine breath test detects significant fibrosis in patients with nonalcoholic steatohepatitis. J Clin Gastroenterol. 2008; 42:408–412
[Reference X2] Fierbinteanu-Braticevici C, Plesca DA, Tribus L, Panaitescu E, Braticevici B. The role of 13C-methacetin breath test for the non-invasive evaluation of nonalcoholic fatty liver disease. J. Gastrointest. Liver Dis. 2013; 22:149–156.
Reviewer 3 Report
This research is incredible valuable for the study of NAFLD. It has significant results that may be introduced into clinical practice. The method is extremely thorough, the results are well presented in tables and figures. The authors discuss their findings in a highly scientific way.
In the methodology section lines 43-44, in regards with alcohol consumption, did you use a validated questionnaire for alcohol consumption (eg. AUDIT)? if yes, please mention. If no, please give some detail on how the amount of alcohol was approximated.
I would suggest including in the discussion section a paragraph indicating the future research that may grow from this reasearch.
Reviewer 4 Report
An interesting study in a large, with liver biopsy defined NAFLD population, that increases the knowledge about the use of the microsomal hepatocellular function.
Introduction
The ABT is widely studied and reviewed in different studies. The metabolic mechanism depends on the function of different P450 cytochromes. Which mono-oxygenase P450 cytochromes are involved in the breakdown of aminopyrine and which factors could disturb its function? This is not explained in the introduction and should be added for a better understanding and fits in the Biochemistry background.
Portal blood flow seems, according to the used reference 5 from 1994, no important factor. Are there no recent studies available because already microcircular changes occur early in fibrogenesis and could have an effect in the speed of metabolism. Liver metabolic capacity, as a dynamic process, maintain stable long and will probably adapt to the anatomical changes, or are there enough scientific arguments against this suggestion?
Methodology section 2.4
With what was aminopyrine ingested, water? orange juice? and how is the absorption? How much is available for metabolism, in other words, what factors could be involved in absorption?
Discussion
The first section is ended with the sentence” the ABT can be….in the appropriate setting.” What is the meaning of the “appropriate setting”, it is rather cryptically formulated.
In different parts of the discussion section fibrosis is the central topic. For the readability these parts should be put together and shortened because it will make the text easier to read.
In the section about serum biomarkers is correctly stated that the majority of biomarkers fail accuracy. What is in this context the added value of ABT as it is functional liver test that reflects a dynamic metabolic process? How accurate is the test to diagnose the different stages, what are the cut off values for the different stages? What are positive and negative predictive values and can it be used as diagnostic noninvasive method? Is this method to be used in combination with ALAT and USS to improve diagnostic accuracy?
What could be the explanation why in simple steatosis the ABT is not compromised? Is it because it dependents on the amount of steatosis? Is there hepatomegaly as a compensatory mechanism to keep the functional capacity of the liver intact in this group or could other factors be involved? Can you speculate more on this aspect?
The section explaining the mechanism in which aminopyrine breakdown is influenced and which factors might be of influence of the metabolism, including ROS, is rather small. It should be elaborated more in depth because understanding the mechanisms of the disturbed dynamic metabolic process is crucial. The most important CYP’s playing a role in aminopyrine metabolism are 2C19, 2C8, 2D6 and 2A1. The sum of the intrinsic clearance of these enzymes are 333, 36, 17 and 13 microliter/min/nmol respectively (Niwa et al xenobiotica 1999). CYP3A does not play a role in the ABT but is mentioned to be changed in NAFLD, however the induction of CYP 2E1 is studied in relation to NASH and alcoholic liver disease but is not measured with the ABT. The CYP involvement in NAFLD should be elaborated in a more in depth way bringing the role of CYP in a broader NAFLD context. What also should be discussed is if the mentioned CYP’s and their polymorphisms could play a role in metabolism and thereby the interpretation of the results, like rapid and slow metabolizer and if this could introduce a bias. Also the concomitant use of medication that could affect metabolism is important because 2C19 and 2D6 are the main CYP’s playing a role in drug metabolism as this is not discussed but should be mentioned for the broader understanding. Was the determination of CYP polymorphisms a consideration for this study? Is smoking influencing all the mentioned aminopyrine CYP metabolizers?
The aspects of the ABT should be clustered in the text related to explanation and factors influencing the test as described above making it more easy to read. The information of other BT is interesting but could be shortened as it brings no specific additional arguments for the ABT.
In the discussion reflection on strong and weak aspects of the study is missing and should be added as is also for the future perspective of this test.
In the conclusion the argumentation of the decision to conduct a liver biopsy and prospective monitoring has not been a point evaluated in the discussion but should be part of the issue diagnostic and prospective monitoring because this is of important for clinician. Is the method also easy to preform or only for specific centers and thereby not available for widespread use as NAFLD is rather common?
Round 2
Reviewer 2 Report
It has been revised in a good way.